# Detecting danger in gridworlds using Gromov's Link Condition

## Abstract

Gridworlds have been long-utilised in AI research, particularly in reinforcement learning, as they provide simple yet scalable models for many real-world applications such as robot navigation, emergent behaviour, and operations research. We initiate a study of gridworlds using the mathematical framework of *reconfigurable systems* and *state complexes* due to Abrams, Ghrist & Peterson. State complexes represent all possible configurations of a system as a single geometric space, thus making them conducive to study using geometric, topological, or combinatorial methods. The main contribution of this work is a modification to the original Abrams, Ghrist & Peterson setup which we introduce to capture agent braiding and thereby more naturally represent the topology of gridworlds. With this modification, the state complexes may exhibit geometric defects (failure of *Gromov's Link Condition*). Serendipitously, we discover these failures occur exactly where undesirable or dangerous states appear in the gridworld. Our results therefore provide a novel method for seeking guaranteed safety limitations in discrete task environments with single or multiple agents, and offer useful safety information (in geometric and topological forms) for incorporation in or analysis of machine learning systems. More broadly, our work introduces tools from geometric group theory and combinatorics to the AI community and demonstrates a proof-of-concept for this geometric viewpoint of the task domain through the example of simple gridworld environments.

## 1   Introduction

The notion of a state (or configuration/phase) space is commonly used in mathematics and physics to represent all the possible states of a given system as a single geometric (or topological) object. This perspective provides a bridge which allows for tools from geometry and topology to be applied to the system of concern. Moreover, certain features of a given system are reflected by some geometric aspects of the associated state space (such as gravitational force being captured by *curvature* in spacetime). Thus, insights into the structure of the original system can be gleaned by reformulating them in geometric terms.

In discrete settings, state spaces are typically represented by graphs or their higher dimensional analogues such as simplicial complexes or cube complexes. Abrams, Ghrist & Peterson's *state complexes* [AG04, GP07] provide a general framework for representing discrete reconfigurable systems as non-positively curved (NPC) cube complexes, giving access to a wealth of mathematical and computational benefits via efficient optimisation algorithms guided by geometric insight [AOS12]. These have been used to develop efficient algorithms for robotic motion planning [ABY14, ABCG17] and self-reconfiguration of modular robots [LR10]. NPC cube complexes also possess rich hyperplane structures which geometrically capture binary classification [CN05, Wis12, Sag14]. However, their broader utility to fields like artificial intelligence (AI) has until now been relatively unexplored.

Our main contribution is the first application of this geometric approach (of using state complexes) to the setting of multi-agent gridworlds. We introduce a natural modification to the state complex appropriate to the setting of gridworlds (to capture the braiding or relative movements of agents); however, this can lead to state complexes which are no longer NPC. Nevertheless, by applying Gromov's Link Condition, we completely characterise when positive curvature occurs in our new state complexes, and relate this to features of the gridworlds (see Theorem 5.2). Serendipitously, we discover that the states where Gromov's Link Condition fails are those in which agents can potentially collide. In other words, collision-detection is naturally embedded into the intrinsic geometry of the system. Current approaches to collision-detection and navigation during multi-agent navigation often rely on modelling and predicting collisions based on large training datasets [KFGE19, FLLP20, QZC$^+$21] or by explicitly modelling physical movements [KIU21]. However, our approach is purely geometric, requires no training, and can accommodate many conceivable types of actions and inter-actions, not just simple movements.

Our work relates to a growing body of research aimed towards understanding, from a geometric perspective, how deep learning methods transform input data into decisions, memories, or actions [HR17, LAG$^+$20, SPG$^+$21, AVBP21, SMK11]. However, such studies do not usually incorporate the geometry of the originating domain or task in a substantial way, before applying or investigating the performance of learning algorithms – and even fewer do so for multi-agent systems. One possible reason for this is a lack of known suitable tools. Our experimental and theoretical results show there is a wealth of geometric information available in (even very simple) task domains, which is accessible using tools from geometric group theory and combinatorics.

# 2 State complex of a gridworld

A *gridworld* is a two-dimensional, flat array of *cells* arranged in a grid, much like a chess or checker board. Each cell can be occupied or unoccupied. A cell may be occupied, in our setting, by one and only one freely-moving agent or movable object. Other gridworlds may include rewards, punishments, buttons, doors, locks, keys, checkpoints, dropbears, etc., much like many basic video games. Gridworlds have been a long-utilised setting in AI research, particularly reinforcement learning, since they are simple yet scalable in size and sophistication [DSHLKT20, WKK20]. They also offer clear analogies to many real-world applications or questions, such as robot navigation [HHA21], emergent behaviour [KAP20], and operations research [LSS$^+$21]. For these reasons, gridworlds have also been developed for formally specifying problems in AI safety [LMK$^+$17].

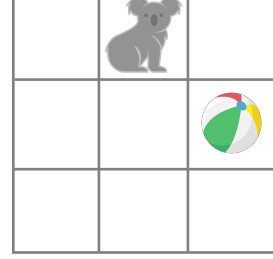

Figure 1: A $3 \times 3$ gridworld with one agent (a koala) and one object (a beach ball).

A *state* of a gridworld can be encoded by assigning each cell a *label*. In the example shown in Figure 1, these labels are shown for an agent, an object, and empty floor. A change in the state, such as an agent moving from one cell to an adjacent empty cell, can be encoded by *relabelling* the cells involved. This perspective allows us to take advantage of the notion of *reconfigurable systems* as introduced by Abrams, Ghrist & Peterson [AG04, GP07].

More formally, consider a graph $G$ and a set $\mathcal{A}$ of labels. A *state* is a function $s : V(G) \to \mathcal{A}$, i.e. an assignment of a label to each vertex of $G$. A possible relabelling is encoded using a *generator* $\phi$; this comprises the following data:

- a subgraph $SUP(\phi) \subseteq G$ called the *support*;
- a subgraph $TR(\phi) \subseteq SUP(\phi)$ called the *trace*; and
- an unordered pair of *local states*

$$u_0^{loc}, u_1^{loc} : V(SUP(\phi)) \to \mathcal{A}$$

that agree on $V(SUP(\phi)) - V(TR(\phi))$ but differ on $V(TR(\phi))$.

A generator $\phi$ is *admissible* at a state $s$ if $s|_{SUP(\phi)} = u_0^{loc}$ (or $u_1^{loc}$), in other words, if the assignment of labels to $V(SUP(\phi))$ given by $s$ completely matches the labelling from (exactly) one of the two

local states. If this holds, we may apply $\phi$ to the state $s$ to obtain a new state $\phi[s]$ given by

$$\phi[s](v) := \begin{cases} u_1^{loc}(v), & v \in V(TR(\phi)) \\ s(v), & \text{otherwise.} \end{cases}$$

This has the effect of relabelling the vertices in (and only in) $TR(\phi)$ to match the other local state of $\phi$. Since the local states are unordered, if $\phi$ is admissible at $s$ then it is also admissible at $\phi[s]$; moreover, $\phi[\phi[s]] = s$.

**Definition 2.1** (Reconfigurable system [AG04, GP07]). A *reconfigurable system* on a graph $G$ with a set of labels $\mathcal{A}$ consists of a set of generators together with a set of states closed under the action of admissible generators.

Configurations and their reconfigurations can be used to construct a *state graph* (or transition graph), which represents all possible states and transitions between these states in a reconfigurable system. More formally:

**Definition 2.2** (State graph). The state graph $\mathcal{S}^{(1)}$ associated to a reconfigurable system has as its vertices the set of all states, with edges connecting pairs of states differing by a single generator.

Let us now return our attention to gridworlds. We define a graph $G$ to have vertices corresponding to the cells of a gridworld, with two vertices declared adjacent in $G$ exactly when they correspond to neighbouring cells (i.e. they share a common side). Our set of labels is chosen to be

$$\mathcal{A} = \{\text{`agent'}, \text{`object'}, \text{`floor'}\}.$$

We do not distinguish between multiple instances of the same label. We consider two generators:

- **Push/Pull.** An agent adjacent to an object is allowed to push/pull the object if there is an unoccupied floor cell straight in front of the object/straight behind the agent; and

- **Move.** An agent is allowed to move to a neighbouring unoccupied floor cell.

These two generators have the effect of enabling agents to at any time move in any direction not blocked by objects or other agents, and for agents to push or pull objects within the environment into any configuration if there is sufficient room to move. For both types of generators, the trace coincides with the support. For the Push/Pull generator, the support is a row or column of three contiguous cells, whereas for the Move generator, the support is a pair of neighbouring cells. A simple example of a state graph, together with the local states for the two generator types, is shown in Figure 2.

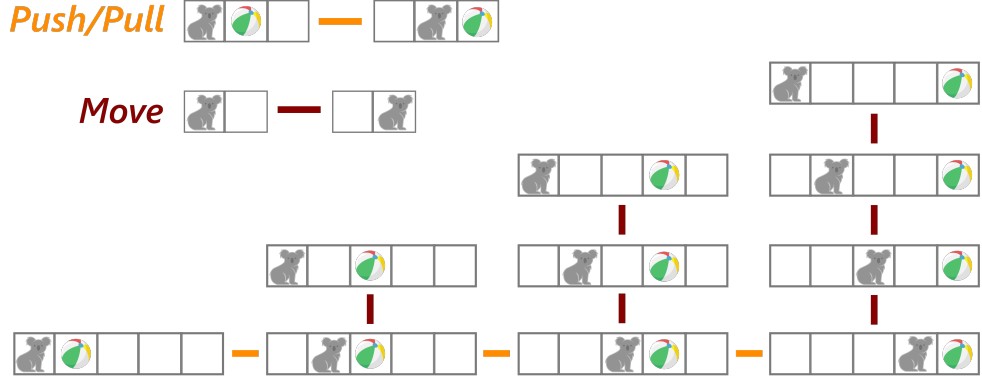

Figure 2: An example $1 \times 5$ gridworld with one agent and one object with two generators – Push/Pull and Move – and the resulting state graph. In the state graph, edge colours indicate the generator type which relabels the gridworld.

In a typical reconfigurable system, there may be many admissible generators at a given state $s$. If the trace of an admissible generator $\phi_1$ is disjoint from the support of another admissible generator $\phi_2$, then $\phi_2$ remains admissible at $\phi_1[s]$. This is because the relabelling by $\phi_1$ does not interfere with

the labels on $SUP(\phi_2)$. More generally, a set of admissible generators $\{\phi_1, \ldots, \phi_n\}$ at a state $s$ *commutes* if $SUP(\phi_i) \cap TR(\phi_j) = \emptyset$ for all $i \neq j$. When this holds, these generators can be applied independently of one another, and the resulting state does not depend on the order in which they are applied. A simple example of this in the context of gridworlds is a large room with $n$ agents spread sufficiently far apart to allow for independent simultaneous movement.

Abrams, Ghrist & Peterson represent this mutual commutativity by adding higher dimensional cubes to the state graph to form a cube complex called the *state complex*. We give an informal definition here, and refer to their papers for the precise formulation [AG04, GP07]. Further background on cube complexes can be found in [Wis12, Sag14]. If $\{\phi_1, \ldots, \phi_n\}$ is a set of commuting admissible generators at a state $s$ then there are $2^n$ states that can be obtained by applying any subset of these generators to $s$. These $2^n$ states form the vertices of an $n$–cube in the state complex. Each $n$–cube is bounded by $2n$ faces, where each face is an $(n-1)$–cube: by disallowing a generator $\phi_i$, we obtain a pair of faces corresponding to those states (in the given $n$–cube) that agree with one of the two respective local states of $\phi_i$ on $SUP(\phi_i)$.

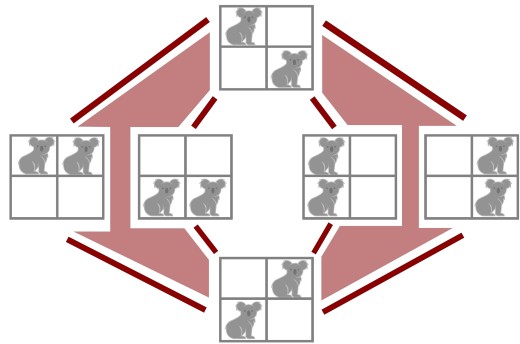

Figure 3: State complex of a $2 \times 2$ gridworld with two agents. Shading indicates squares attached to the surrounding 4–cycles.

**Definition 2.3** (State complex). The *state complex* $\mathcal{S}$ of a reconfigurable system is the cube complex constructed from the state graph $\mathcal{S}^{(1)}$ by inductively adding cubes as follows: whenever there is a set of $2^n$ states related by a set of $n$ admissible commuting generators, we add an $n$–cube so that its vertices correspond to the given states, and so that its $2n$ boundary faces are identified with all the possible $(n-1)$–cubes obtained by disallowing a generator. In particular, every cube is uniquely determined by its vertices.

In our gridworlds setting, each generator involves exactly one agent. This means commuting generators can only occur if there are multiple agents. A simple example of a state complex for two agents in a $2 \times 2$ room is shown in Figure 3. Note that there are six embedded 4–cycles in the state graph, however, only two of these are filled in by squares: these correspond to independent movements of the agents, either both horizontally or both vertically.

## 3 Exploring gridworlds with state complexes

To compute the state complex of a (finite) gridworld, we first initialise an empty graph $\mathcal{G}$ and an empty 'to-do' list $\mathcal{L}$. As input, we take a chosen state of the gridworld to form the first vertex of $\mathcal{G}$ and also the first entry on $\mathcal{L}$. The state complex is computed according to a breadth-first search by repeatedly applying the following:

- Let $v$ be the first entry on $\mathcal{L}$. List all admissible generators at $v$. For each such generator $\phi$:
    - If $\phi[v]$ already appears as a vertex of $\mathcal{G}$, add an edge between $v$ and $\phi[v]$ (if it does not already exist).
    - If $\phi[v]$ does not appear in $\mathcal{G}$, add it as a new vertex to $\mathcal{G}$ and add an edge connecting it to $v$. Append $\phi[v]$ to the end of $\mathcal{L}$.
- Remove $v$ from $\mathcal{L}$.

The process terminates when $\mathcal{L}$ is empty. The output is the graph $\mathcal{G}$. When $\mathcal{L}$ is empty, we have fully explored all possible states that can be reached from the initial state. It may be possible that the true state graph is disconnected, in which case the above algorithm will only return a connected component $\mathcal{G}$. For our purposes, we shall limit our study to systems with connected state graphs. From the state graph, we construct the state complex by first finding all 4–cycles in the state graph. Then, by examining the states involved, we can determine whether a given 4–cycle bounds a square representing a pair of commuting moves.

To visualise the state complex, we first draw the state graph using the Kamada–Kawai force-directed algorithm [KK89] which attempts to draw edges to have similar length. We then shade the region(s) enclosed by 4–cycles representing commuting moves. For ease of visual interpretation in our figures, we do not also shade higher-dimensional cubes, although such cubes are noticeable and can be easily computed and visualised if desired.

Constructing and analysing state complexes of gridworlds is in and of itself an interesting and useful way of exploring their intrinsic geometry. For example, Figure 4 shows the state complex of a $3 \times 3$ gridworld with one agent and one object. The state complex reveals two scales of geometry: larger 'blobs' of states organised in a $3 \times 3$ grid, representing the location of the object; and, within each blob, copies of the room's remaining empty space, in which the agent may walk around and approach the object to Push/Pull. Each 12–cycle 'petal' represents a 12–step choreography wherein the agent pushes and pulls the object around in a 4–cycle in the gridworld. In this example, the state complex is the state graph, since there are no possible commuting moves.

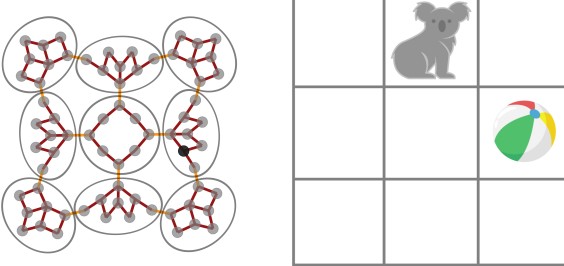

Figure 4: State complex (left) of a $3 \times 3$ gridworld with one agent and one object (right). The darker vertex in the state complex represents the state shown in the gridworld state on the right. Edges in the state complex are coloured according to their generator – orange for Push/Pull and maroon for Move. Grey circles which group states where the ball is static have been added to illustrate the different scales of geometry.

The examples discussed thus far all have planar state graphs. Planarity does not hold in general – indeed, the $n$–cube graph for $n \geq 4$ is non-planar, and a state graph can contain $n$–cubes if the gridworld has $n$ agents and sufficient space to move around. It is tempting to think that the state complex of a gridworld with more agents should therefore look quite different to one with fewer agents. However, Figure 5 shows this may not always be the case: there is a symmetry induced by swapping all 'agent' labels with 'floor' labels.

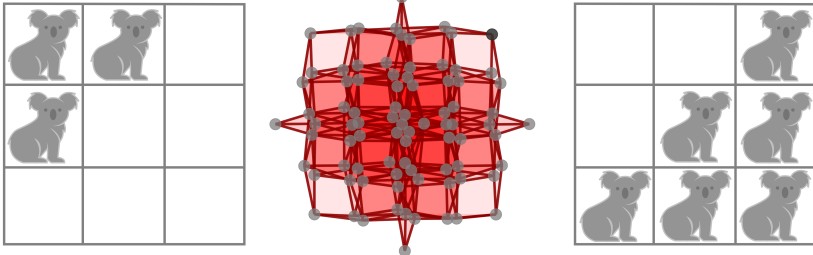

Figure 5: State complex (centre) of a $3 \times 3$ gridworld with three agents (left) and six agents (right). They share the same state complex due to the 'agent' $\leftrightarrow$ 'floor' label inversion symmetry.

## 4 Dancing with myself

The state complex of a gridworld with $n$ agents can be thought of as a discrete analogue of the configuration space of $n$ points on the 2D–plane. However, there is a problem with this analogy: there can be 'holes' created by 4–cycles in the state complex where a single agent walks in a small square-shaped dance by itself, as shown in Figure 6.

The presence of these holes would suggest something meaningful about the underlying gridworld's intrinsic topology, e.g., something obstructing the agent's movement at that location in the gridworld that the agent must move around. In reality, the environment is essentially a (discretised) 2D–plane with nothing blocking the agent from traversing those locations. Indeed, these 'holes' are uninteresting

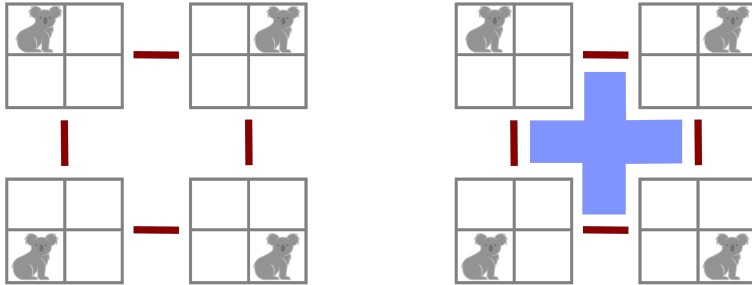

Figure 6: State complex of a $2 \times 2$ gridworld with one agent under the original definition of Abrams, Ghrist & Peterson [AG04, GP07] (left) and with our modification (right). The blue shading is a filled in square indicating a *dance*.

topological quirks which arise due to the representation of the gridworld as a graph. We therefore deviate from the original definition of state complexes by Abrams, Ghrist & Peterson [AG04, GP07] and choose to fill in these 'dance' 4–cycles with squares.[1]

Formally, we define a **dance** $\delta$ to comprise the following data:

- the support $SUP(\delta)$ given by a $2 \times 2$ subgrid in the gridworld,

- four local states defined on $SUP(\delta)$, each consisting of exactly one agent label and three floor labels, and

- four Move generators, each of which transitions between two of the four local states (as in Figure 6).

We say that $\delta$ is *admissible* at a state $s$ if $s|_{SUP(\delta)}$ agrees with one of the four local states of $\delta$. Moreover, these four local states are precisely the states that can be reached when we apply some combination of the four constituent Moves. We do not define the trace of a dance, however, we may view the trace of each of the four constituent Moves as subgraphs of $SUP(\delta)$.

The notion of commutativity can be extended to incorporate dancing. Suppose that we have a set $\{\phi_1, \ldots, \phi_l, \delta_1, \ldots, \delta_m\}$ of $l$ admissible generators and $m$ admissible dances at a state $s$. We say that this set *commutes* if the supports of its elements are pairwise disjoint. When this holds, there are $2^{l+2m}$ possible states that can be obtained by applying some combination of the generators and dances to $s$: there are two choices of local state for each $\phi_i$, and four for each $\delta_j$. We capture this extended notion of commutativity by attaching additional cubes to the state complex to form our modified state complex.

**Definition 4.1** (Modified state complex). The *modified state complex* $\mathcal{S}'$ of a gridworld is the cube complex obtained by filling in the state graph $\mathcal{S}^{(1)}$ with higher dimensional cubes whenever there is a set of commuting moves or dances. Specifically, whenever a set of $2^{l+2m}$ states are related by a commuting set of $l$ generators and $m$ dances, we add an $n$–cube having the given set of states as its vertices, where $n = l + 2m$. Each of the $2n$ faces of such an $n$–cube is identified with an $(n-1)$–cube obtained by either disallowing a generator $\phi_i$ and choosing one of its two local states, or replacing a dance $\delta_j$ with one of its four constituent Moves.

Our modification removes uninteresting topology. This can be observed by examining 4–cycles in $\mathcal{S}'$. On the one hand, some 4–cycles are trivial (they can be 'filled in'): *dancing-with-myself* 4–cycles, and *commuting moves* (two agents moving back and forth) 4–cycles (which were trivial under the original definition). These represent trivial movements of agents relative to one another. On the other hand, there is a non-trivial 4–cycle in the state complex for two agents in a $2 \times 2$ room, as can be seen in the centre of Figure 3 (here, no dancing is possible so the modified state complex is the same as the original). This 4–cycle represents the two agents moving half a 'revolution' relative to one another –

---

[1]Ghrist and Peterson themselves ask if there could be better ways to complete the state graph to a higher-dimensional object with better properties (Question 6.4 in [GP07]).

indeed, performing this twice would give a full revolution. (There are three other non-trivial 4–cycles, topologically equivalent to this central one, that also achieve the half-revolution.)

In a more topological sense[2], by filling in such squares and higher dimensional cubes, our state complexes capture the non-trivial, essential relative movements of the agents. This can be used to study the braiding or mixing of agents, and also allows us to consider path-homotopic paths as 'essentially' the same. One immediate difference this creates with the original state complexes is a loss of symmetries like those shown in Figure 5, since there is no label inversion for a dance when other agents are crowding the dance-floor.

## 5 Gromov's Link Condition

The central geometric characteristic of Abrams, Ghrist, & Peterson's state complexes is that they are *non-positively curved* (NPC). Indeed, this local geometric condition is conducive for developing efficient algorithms for computing geodesics. However, with our modified state complexes, this NPC geometry is no longer guaranteed – we test for this on a vertex-by-vertex basis using a classical geometric result due to Gromov (see also Theorem 5.20 of [BH99] and [Sag14]).

**Theorem 5.1** (Gromov's Link Condition [Gro87])**.** *A finite-dimensional cube complex is NPC if and only if the link of every vertex is a flag simplicial complex.* $\qquad\square$

We provide a brief mathematical background on cube complexes and the finer details of Gromov's Link Condition in Appendix A.1. For our current purposes, it is sufficient to know that under the Abrams, Ghrist & Peterson setup, if $v$ is a state in $\mathcal{S}$ then the vertices of its link $lk(v)$ represent the possible admissible generators at $v$. Since cubes in $\mathcal{S}$ are associated with commuting sets of generators, each simplex in $lk(v)$ represents a set of commuting generators. Gromov's Link Condition for $lk(v)$ can be reinterpreted as follows: whenever a set of admissible generators is *pairwise* commutative, then it is *setwise* commutative. Using this, it is straightforward for Abrams, Ghrist & Peterson to verify that this always holds for their state complexes (see Theorem 4.4 of [GP07]).

For our modified states complexes, the situation is not as straightforward. The key issue is that our cubes do not only arise from commuting generators – we must take dances into account. Indeed, when attempting to prove that Gromov's Link Condition holds, we discovered some very simple gridworlds where it actually fails; see Figure 7 and Appendix A.4.

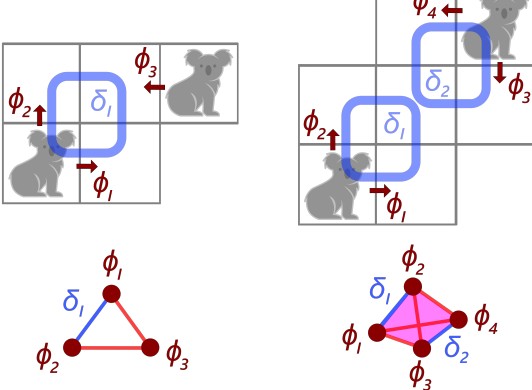

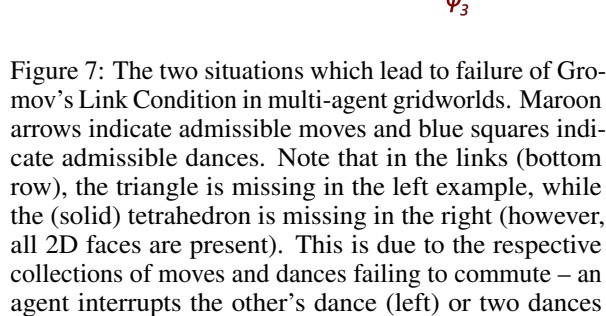

Figure 7: The two situations which lead to failure of Gromov's Link Condition in multi-agent gridworlds. Maroon arrows indicate admissible moves and blue squares indicate admissible dances. Note that in the links (bottom row), the triangle is missing in the left example, while the (solid) tetrahedron is missing in the right (however, all 2D faces are present). This is due to the respective collections of moves and dances failing to commute – an agent interrupts the other's dance (left) or two dances collide (right).

Failure of the Link Condition can indicate available moves at some state that cannot be safely performed simultaneously and independently without risking collisions between labels. Another interpretation of positive curvature in this context is something akin to what real-time computer strategy games call 'fog of war' (distance-dependent limiting of observations which extends from the player-controlled agents), and more specifically the viewable distance from an agent's line-of-sight. Such fog makes AI systems operating in such environments particularly challenging, although remarkable success has been achieved in games like StarCraft [VBC+19].

---

[2]By considering the fundamental group.

Despite this apparent drawback, we nevertheless show that Figure 7 accounts for all the possible failures of Gromov's Link Condition in the setting of agent-only gridworlds[3].

**Theorem 5.2** (Gromov's Link Condition in the modified state complex). *Let $v$ be a vertex in the modified state complex $\mathcal{S}'$ of an agent-only gridworld. Then*

- *$lk(v)$ satisfies Gromov's Link Condition if and only if it has no empty 2–simplices nor 3–simplices[4], and*

- *if $lk(v)$ fails Gromov's Link Condition then there exist a pair of agents whose positions differ by either a knight move or a 2–step bishop move (as in Figure 7).*

We provide a proof in Appendix A.2. Consequently, if the Link Condition fails at all, it must fail at dimension 2 or 3. This can be interpreted as saying that we only need a bounded amount of foresight to detect potential collisions: under fog-of-war, each agent needs a line-of-sight of only four moves.

Positive curvature could indicate collisions between any specified labels (e.g., objects), however, for this interpretation to be valid we would need to carefully identify which other potential cycles in the state complex ought to be filled in. Doing this in a 'natural' way is in itself a non-trivial task, and is the subject of further investigation.

# 6   Experiments and applications

Although our main contribution is theoretical, we conduct some small initial experiments to demonstrate the type of information which can be captured in the geometry and topology (see Appendix A.4). To run these experiments, we developed and used a custom Python-based tool (detailed in Appendix A.3). Our focus on small rooms is largely expository, i.e., they are the simplest non-trivial examples illustrating the key features we want to isolate, and naturally reoccur in all larger rooms. Our intention is also to demonstrate a combinatorial explosion in the number of states. We don't recommend constructing the entire state complex in practical applications (indeed, to implement addition of integers on a computer, it is infeasible and unnecessary to construct *all* integers).

*Remark* 6.1. By a simple counting argument, one can deduce the total number of states in a gridworld. For an agent-only gridworld with $n$ cells and $k$ agents, there is a total of $\binom{n}{k}$ states. If there are $n$ cells, $k$ agents, and $j$ objects, then there are $\binom{n}{k}\binom{n-k}{j}$ states. Thus, even for a moderately sized $10 \times 10$ room with 50 agents, there are $\binom{100}{50} \approx 1.008 \times 10^{29}$ vertices in the state complex.

By Theorem 5.2, checking if $lk(v)$ satisfies Gromov's Link Condition requires computing the link only up to dimension 3 and then checking whether it is a flag complex; if not, we count the number of empty simplices. Checking this for a given vertex in the state complex is not too computationally demanding, however when a state complex has many vertices it becomes more difficult. In practical applications, such as calculating collision-avoiding navigation routes, it is – again, by Theorem 5.2 – only necessary to construct a small local subcomplex. But perhaps even more importantly, to detect potential collisions between agents, it is not even necessary to construct $lk(v)$, since Theorem 5.2 provides a computational shortcut: just check for supports of knight or two-step bishop moves between agents.

By using Gromov's Link Condition, we can identify a precise measure of how far ahead agents ought to look in order to safely proceed without fear of collisions. Appendix A.4 gives a summary analysis of a $3 \times 3$ room with varying numbers of agents. We noticed several symmetries. Commuting moves and the number of states have a symmetry about $4.5$ agents (due to the label-inversion symmetry as previously illustrated in Figure 5). However, curiously, the number of dances has a symmetry about $3.5$ agents. This difference leads to the asymmetrical distribution of positive curvature and failures of Gromov's Link Condition – which, while maximal for 3 agents as a proportion of total states, exhibited the highest mean failure rate for 4 agents.

---

[3]While writing this paper, the first author was involved in two scooter accidents – collisions involving only agents (luckily without serious injury). So, while this class of gridworlds is strictly smaller than those also involving objects or other labels, it is by no means an unimportant one. If only the scooters had Gromov's Link Condition checkers!

[4]In other words, if there are no "hollow" triangles or tetrahedra like those in Figure 7.

This shows that, heuristically, we expect most states to satisfy NPC (see Appendix A.4), and so existing greedy algorithms [AOS12] for calculating geodesics will work well in most situations. However, to implement an efficient, collision-free path-finding algorithm in our modified state complexes, we need to add an additional check. Specifically, when we are near a potentially dangerous state, we should implement a predefined 'detour' to avoid the collision, which can be done on a local basis using the identified supports which lead to positive curvature (as in Figure 7).

# 7   Conclusions and future directions

This study presents novel applications of tools from geometric group theory and combinatorics to the AI research community, opening new ways for recasting and analysing AI problems as geometric ones. Using these tools, we show an example of how the intrinsic geometry of a task space serendipitously embeds safety information and makes it possible to determine how far ahead in time an AI system needs to observe to be guaranteed of avoiding dangerous actions.

Leike et al. [LMK+17] show deep reinforcement learning agents cannot solve many AI safety problems specified on gridworlds, e.g., minimising unwanted side-effects or ensuring robustness to agent self-modification. Having described the agent-only case in this study, there is now ripe opportunity to account for positive curvature or other geometric features arising due to other labels or generators (actions) present in specified AI safety problems, e.g., agents pushing/pulling objects, pressing buttons, modifying their form or behaviour, rewards/punishments, opening/unlocking doors, etc.. By considering *directed* modified state complexes, irreversible actions can be captured by "invariant subcomplexes" (i.e., you can't escape from them), allowing geometric study of the tree/flowchart of irreversible actions and related recurrence/transience. Braiding can be used to study route planning, back-tracking, cooperation, assembly, and topological entropy in congestion [Ghr09]. Numerous extensions are possible, allowing us to study and geometrically represent further problems with a view to developing efficient, geometrically-inspired local algorithms without the need for training.

Do learning algorithms already implement such geometrically-inspired algorithms, the related geometry, or approximations thereof? To find out, we are investigating how modified state complexes map to learned internal representations of neural networks trained to predict multi-agent gridworld dynamics. This mapping connects the geometry and topology of a task space directly to optimisation procedures and learning trajectories in latent representation spaces, highlighting unexpected topological and geometric differences and opportunities for deeper insight and improvement of optimisation procedures, in the spirit of [NZL20, ZZ22]. We can also compare biological optimisation processes and internal representations of allocentric and egocentric navigation [Bur06, GHP+22], and how this interacts with the position of other agents [DJ18, SB20].

From a more mathematical perspective, state complexes of gridworlds give rise to an interesting class of geometric spaces. It would be worthwhile to investigate their geometric and topological properties to more deeply understand various aspects of multi-agent gridworlds. For example, for a gridworld with $n$ agents in a sufficiently large room, we hypothesise that the modified state complex should be a classifying space for the $n$–strand braid group. This is clearly false when the room is packed full of agents (in which case the state complex is a single point), so it may be fruitful to determine if there is some 'critical' density at which a topological transition occurs.

Using the *failure* of Gromov's Link Condition in an essential way appears to be a relatively unexplored approach. Indeed, much of the mathematical literature concerning cube complexes focusses on showing that the Link Condition always holds. To our knowledge, the only other works which go against this trend are [AG04], in which failure detects global disconnection of a metamorphic system, and [BDT19], where failure detects non-trivial loops on topological surfaces. It would be interesting to explore cube complexes arising in other settings where failure captures critical information.

A limitation of our work is that we have so far only explored very simple AI environments. Further work is needed to expand the framework and results to more general, sophisticated, and real-world environments. For this reason, although our work provides new geometric perspectives, data, and potential algorithms for an important AI safety issue, we caution against hasty real-world implementation of the main results. To avoid potential negative societal impacts, it would still be important to perform rigorous checks and tests in application domains, since our results do not directly extend to situations beyond which the stated assumptions hold.

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

| 449 | [LMK⁺17] | Jan Leike, Miljan Martic, Victoria Krakovna, Pedro A. Ortega, Tom Everitt, Andrew Lefrancq, Laurent Orseau, and Shane Legg, *AI safety gridworlds*, arXiv:1711.09883, 2017. |

[LMK$^+$17]  Jan Leike, Miljan Martic, Victoria Krakovna, Pedro A. Ortega, Tom Everitt, Andrew Lefrancq, Laurent Orseau, and Shane Legg, *AI safety gridworlds*, arXiv:1711.09883, 2017.

[LR10]  Tom Larkworthy and Subramanian Ramamoorthy, *An efficient algorithm for self-reconfiguration planning in a modular robot*, 2010 IEEE International Conference on Robotics and Automation, 2010, pp. 5139–5146.

[LSS$^+$21]  Florian Laurent, Manuel Schneider, Christian Scheller, Jeremy Watson, Jiaoyang Li, Zhe Chen, Yi Zheng, Shao-Hung Chan, Konstantin Makhnev, Oleg Svidchenko, Vladimir Egorov, Dmitry Ivanov, Aleksei Shpilman, Evgenija Spirovska, Oliver Tanevski, Aleksandar Nikov, Ramon Grunder, David Galevski, Jakov Mitrovski, and Sharada Mohanty, *Flatland competition 2020: MAPF and MARL for efficient train coordination on a grid world*, pp. 275–301, PMLR, 08 2021.

[NZL20]  Gregory Naitzat, Andrey Zhitnikov, and Lek-Heng Lim, *Topology of deep neural networks*, Journal of Machine Learning Research **21** (2020), no. 184, 1–40.

[QZC$^+$21]  Zengyi Qin, Kaiqing Zhang, Yuxiao Chen, Jingkai Chen, and Chuchu Fan, *Learning safe multi-agent control with decentralized neural barrier certificates*, International Conference on Learning Representations, 2021.

[Sag14]  Michah Sageev, $CAT(0)$ *cube complexes and groups*, Geometric group theory, IAS/Park City Math. Ser., vol. 21, Amer. Math. Soc., Providence, RI, 2014, pp. 7–54.

[SB20]  Christina J. Sutherland and David K. Bilkey, *Hippocampal coding of conspecific position*, Brain Research **1745** (2020), 146920.

[SMK11]  Jeremy Stober, Risto Miikkulainen, and Benjamin Kuipers, *Learning geometry from sensorimotor experience*, 2011 IEEE International Conference on Development and Learning (ICDL), vol. 2, 2011, pp. 1–6.

[SPG$^+$21]  Cory Stephenson, Suchismita Padhy, Abhinav Ganesh, Yue Hui, Hanlin Tang, and SueYeon Chung, *On the geometry of generalization and memorization in deep neural networks*, International Conference on Learning Representations, 2021.

[VBC$^+$19]  Oriol Vinyals, Igor Babuschkin, Wojciech M Czarnecki, Michaël Mathieu, Andrew Dudzik, Junyoung Chung, David H Choi, Richard Powell, Timo Ewalds, Petko Georgiev, Junhyuk Oh, Dan Horgan, Manuel Kroiss, Ivo Danihelka, Aja Huang, Laurent Sifre, Trevor Cai, John P Agapiou, Max Jaderberg, Alexander S Vezhnevets, Rémi Leblond, Tobias Pohlen, Valentin Dalibard, David Budden, Yury Sulsky, James Molloy, Tom L Paine, Caglar Gulcehre, Ziyu Wang, Tobias Pfaff, Yuhuai Wu, Roman Ring, Dani Yogatama, Dario Wünsch, Katrina McKinney, Oliver Smith, Tom Schaul, Timothy Lillicrap, Koray Kavukcuoglu, Demis Hassabis, Chris Apps, and David Silver, *Grandmaster level in StarCraft II using multi-agent reinforcement learning*, Nature **575** (2019), no. 7782, 350–354.

[Wis12]  Daniel T. Wise, *From riches to raags: 3-manifolds, right-angled Artin groups, and cubical geometry*, CBMS Regional Conference Series in Mathematics, vol. 117, Published for the Conference Board of the Mathematical Sciences, Washington, DC; by the American Mathematical Society, Providence, RI, 2012.

[WKK20]  Vikram Waradpande, Daniel Kudenko, and Megha Khosla, *Deep reinforcement learning with graph-based state representations*, arXiv:2004.13965, 2020.

[ZZ22]  Yang Zhao and Hao Zhang, *Quantitative performance assessment of CNN units via topological entropy calculation*, International Conference on Learning Representations, 2022.

