# OpenReview forum: "Detecting danger in gridworlds using Gromov's Link Condition"
_NeurIPS.cc/2022/Conference — NeurIPS 2022 Submitted_

### Official Review · Reviewer_5vs3 · 2022-07-12

**Rating:** 3
**Confidence:** 4
**Soundness:** 2 fair
**Presentation:** 2 fair
**Contribution:** 2 fair

**Summary:**

The paper studies the geometry of the state space specific to 2-dimensional grid-world environments. The methods provide a way of identifying unsafe states in discrete environments.

**Questions:**

See above.

**Limitations:**

See above.

**Strengths And Weaknesses:**

Although the study of geometric structures of state complexes is interesting, it is hard to see the computational benefits of doing so. In particular, it is hard for me to see where the geometry of state transitions can help with learning beyond standard RL. In particular, the identification of unsafe states by observing transitions between states can be exactly captured by Q-learning, which propagates values to implicitly identify the geometry of the state complexes.

The experiments have not attempted to compare with any RL baselines, while the complexities of Q-learning or value iteration can be directly analyzed in the simple gridworld setting.

---

> ### Author Response · Authors · 2022-08-01
> **Response to Reviewer 5vs3**
>
> **State complexes vs. transition graphs**
>
> We understand that techniques such as Q-learning can be useful for assigning values to actions taken on transition graphs, and that such transition graphs can even be built explicitly alongside such techniques to improve performance, e.g., as a data-efficient back-up [JZK+22]. However, we are unaware of any literature showing this for higher-dimensional complexes – and, indeed, we believe we are the first to introduce such complexes to the domain of gridworlds and discrete AI/configuration spaces. If we are mistaken about this, please let us know and provide references for us to look at. We have double-checked the citing article list of relevant Abrams-Ghrist-Peterson papers, which introduced state complexes, and found no such articles (we assume any ML techniques which attempt to recover/build state complexes would cite the originating works).
>
> Should prior literature exist which shows how Q-learning can exactly recover higher dimensional cubes, we would be very interested to know precisely how such techniques determine which higher-dimensional cells ought to be attached. This is highly non-trivial, as there are typically many ways in which a graph can be extended to a higher-dimensional complex, e.g., for the transition graph, we could extend it to AGP's state complexes, or to our modified state complexes. It would therefore be interesting to know whether certain methods prefer to fill in cells in a particular way. To the best of our knowledge, such studies are yet to be completed, however we believe they will certainly be interesting to pursue in the future.
>
> **Benefits of this new geometric viewpoint**
>
> More broadly, our paper opens new ways for recasting and analysing AI problems as geometric ones. Notice: we never sought out to capture safety information in the geometric representation, it just serendipitously ‘appeared’ as an embedded geometric property of the task space. We intend to extend this framework to study more sophisticated AI environments and more deeply study our modified state complexes. We believe taking these geometric perspectives will aid in AI transparency/safety efforts, and offer novel algorithmic, learning, and analysis opportunities. E.g., do learning algorithms already implement the type of geometrically-inspired algorithms which perform so efficiently on cube complexes, or approximations thereof? Or the related geometry? What about biological representations of allocentric and egocentric navigation? We hope these questions and many others (like those provided in the Discussion section of the manuscript) will be tackled in the near future.
>
> **Additional references**
>
> [JZK+22] Zhengyao Jiang, Tianjun Zhang, Robert Kirk, Tim Rocktäschel, Edward Grefenstette. Graph Backup: Data Efficient Backup Exploiting Markovian Transition. arXiv:2205.15824 (2022).

---

### Official Review · Reviewer_txoD · 2022-07-12

**Rating:** 4
**Confidence:** 2
**Soundness:** 3 good
**Presentation:** 3 good
**Contribution:** 2 fair

**Summary:**

This paper considers gridworlds, 2D grids where cells are occupied by objects that can either move around the grid themselves or be moved by other objects. The authors use state complexes to analyse multi-agent gridworlds, and find that they are not always non-positively curved by determining if Gromov's Link Condition is satisfied or not. The condition fails in situations where two agents might collide with each other, thus the condition can be used as a test for safety in the gridworld. They find that agents separated by either a knight or 2-step bishop move could collide, and interpret this as saying that agents only need to know what other agents are doing 4 steps in advance to avoid potential collisions.

**Questions:**

(1) What does it mean for a square in the state complex to be "filled in", and why is this beneficial? (see "strengths and weaknesses" section for more detail)

(2) The authors acknowledge that constructing the entire state complex of large gridworlds is computationally intractable, so I wonder about how this work could be applied to other areas, as discussed in the Conclusions. Gridworlds seem like a very simple environment to apply these geometric methods to, and you already find that state complexes become too large to work with for even moderately sized environments. How do you think other problems could be tackled, given how computationally expensive the simple environments in this paper are to analyse?

**Limitations:**

I believe limitations have mostly been acknowledged, but have asked for a little more information in question (2).

**Strengths And Weaknesses:**

The main contribution and strength of this paper is in its theoretical developments. A mathematical result is derived in detail, and verified by simulation as well. While the end result (knight and 2-step bishop separations are potentially dangerous, and a 4 step line of sight is sufficient) is not groundbreaking, the methods the authors use to reach these conclusions are interesting and could be applied to other scenarios, as mentioned in the Conclusions.

While I am not an expert in this area, I feel the authors mostly did a good job of explaining the concepts with clear diagrams, so I could gain a good general understanding of the work. However, I'm somewhat confused by what the modification in Section 4 achieves. What effect does "filling in" the holes in the state complex brought about by dances have? I think I'm probably missing what it means for a square to be "filled in", so can this be explained better? Given Sections 2 and 3 seem to basically summarise the work of Abrams, Ghrist, and Peterson whereas Section 4 is the new contribution of this paper, perhaps Sections 2 and 3 could be shortened to allow more space to discuss Section 4.

I found Figure 7 hard to understand initially. It would be beneficial to include a diagram of the state complex to show how the links come about (similar to Figure 8 in Appendix A1). I also don't quite understand what is meant by "the triangle is missing in the left example", as this shows a triangle but with one of the edges in blue instead of red. This relates back to my previous point about what "filled in" squares actually mean: I guess you're trying to point out that the triangle is not filled in. Perhaps you could include an example of a situation in which Gromov's Link Condition *does* hold, in order to better illustrate the differences.

Finally, there are a few examples of informal language that should be removed in any published version of this work. For example, page 2 "dropbears" or footnote 3 on page 8.

Following the rebuttal, the score has been adjusted to take into account all comments by the authors and reviewers.

---

> ### Author Response · Authors · 2022-08-01
> **Response to Reviewer txoD**
>
> **Filling-in cells and Gromov’s Link Condition**
>
> Let us consider two situations: making a single diagonal cut which removes the corner of a cardboard box; and a similar cut which removes a corner from a solid block of cheese. In both cases, the cross-section of these cuts are triangular, however the interior is empty for the box and “filled in” for the cheese. That’s because the box was hollow (geometrically, a collection of square panels glued along their edges) whereas the cheese was solid (geometrically, a 3D cube). Geometrically, hollow cubes (more precisely, hollow corners of cubes) give rise to positive curvature (see Appendix A1).
>
> The link of a vertex $v$ in a cube complex describes its local geometry. Intuitively, $lk(v)$ is the union of the cross-sections of cuts taken over all the corners of cubes incident to $v$ (think of taking an intersection with a small sphere centred $v$). Combinatorially, $lk(v)$ is a simplicial complex (formed by gluing triangles, tetrahedra, etc. together). Gromov’s Link Condition forbids “hollow” simplices in $lk(v)$; in other words, they must all be “solid” or “filled in” (like the cheese cut). Thus, the *failure* of Gromov’s Link Condition indicates the presence of positive curvature (arising from a hollow corner of a cube, possibly of higher dimension).
>
> In state complexes, (filled in) squares and cubes represent sets of commutable actions. As it happens, the original Abrams-Ghrist-Peterson (AGP) set-up creates cube complexes which are non-positively curved (NPC) everywhere. This is easy for them to show because, by construction, AGP only add squares or cubes when actions are commutable. However, as discussed in Section 4, this set-up causes unnatural topology in the case of gridworlds (L201-211). Specifically, when an agent walks around in a 4-cycle there is a hole in the cube complex, i.e. a ‘hollow square’. We therefore attach a square on to this 4-cycle and thereby ‘fill it in’.
>
> Formally, we attach a 2-cell (a topological disc) by gluing its boundary to the 4-cycle (a topological circle). This is a standard topological operation used for constructing cell complexes. The mathematical benefit of doing so is to remove uninteresting/irrelevant topological features. Specifically, the 4-cycle is a non-trivial loop in the hollow square (i.e. it is non-trivial in the fundamental group). However, once we attach a 2-cell, this loop can be continuously contracted to a point, and is thus trivial.
>
> Consequently, the topology of our modified state complexes ignores these 4-step dances. Instead, the topology that remains captures the essential relative movements (i.e. braiding) of the agents. This makes it more suitable as a discrete analogue of a configuration space of the agents.
>
> While our modification improved the topology, it ended up causing defects to the geometry, specifically, the failure of Gromov’s Link Condition. Nevertheless, our main theorem shows that this is related to potential collisions of agents. Thus, serendipitously, our modifications led to two connections for the price of one: topology to relative braiding, and local geometry  to collision-checking.
>
> **Writing**
>
> Although Sections 2 and 3 summarise the work of AGP, it also outlines the first application of state complexes to gridworlds, how to compute and visualise such state complexes, and discusses several simple examples for exposition and to guide intuition. This serves not only as part of our contribution (interpreting gridworlds as an AGP reconfigurable system), but also to introduce readers unfamiliar with AGP’s work to the geometric and topological concepts, structures, and tools which we build upon in later sections. However, there may be particular parts of Sections 2 and 3 which could be shortened or moved to an appendix to improve readability. This might indeed be necessary for space in order to add more discussion and explanations to Section 4, as txoD suggests. For example, some of the text in this reply under “Filling-in cells and Gromov’s Link Condition” could be included in Section 4, perhaps even with a small illustration/figure showing an “experiment” of cutting off a corner of a hollow box vs. a block of cheese.
>
> We are happy to remove the informal language, and to add the state complex to Figure 7.
>
> **Computational limitations**
>
> As discussed in our general comment under “Computational considerations”, there is a misconception that dealing with very large objects makes the problem computationally intractable. This is not always the case, and indeed for collision-checking we showed that only a small subset of the much larger structure is necessary to compute. We believe other AI problems, even those which are more sophisticated, can be tackled via a similar geometric perspective. To achieve this, we will likely need to develop and use other tools from geometric group theory, topology, and geometry, e.g., universal covers, Word Problem in groups, hyperplanes in cube complexes.

---

### Official Review · Reviewer_2SSW · 2022-07-22

**Rating:** 2
**Confidence:** 3
**Soundness:** 2 fair
**Presentation:** 1 poor
**Contribution:** 1 poor

**Summary:**

This paper advocates for the use of group theory to characterize geometric features of grid worlds. The paper claims these geometric features reflect information about obstructions that can ultimately be used to check for collisions. The paper develops its ideas through an extension of state complexes, then formally argues that preconditions for a special geometric property (Gromov’s Link Condition) are not satisfied when some geometric features are present.

Understanding how domain geometry constrains the agency of a decision maker is an ambitious and potentially interesting line of research. However, the current paper has not done enough in this regard to be published, as it fails to communicate how its ideas can be operationalized in AI research. The paper's poor organization is one of its major issues, as is its largely unsubstantiated claims and unconvincing empirical remarks. All taken together, the paper needs more work to achieve the goals it sets for itself. I believe that with a major overhaul this could eventually become an interesting line of research.

**Questions:**

* Consider that AI systems are often used in settings when the full state space topology is unknown. Do you see these ideas being relevant in the absence of that information?
* What is the difference between a state complex and the configuration space used in robotics [(LaVelle, 2006)](http://lavalle.pl/planning/node144.html)?


**Limitations:**

### Significance

The paper centers around the claim that its ideas help with collision checking. However, the notion of collisions here is totally different than the way others conceptualize collisions. The paper seems to view collisions as something gives a state space its inherent topology, even in settings where the topology is just a product of discretization. So even in an empty space, the paper assumes an obstacle prevents the agent from moving diagonally between a four-action topology. This seems strange, and its not obvious this abstraction buys an AI researcher additional insight into the domain.

Computing the objects needed to perform the collision checking seems computationally intractable, even for small domains. The paper even mentions in Remark 6.1 computing a state complex for a 10x10 grid world and multiple agents requires on the order of $10^29$ states. With such a combinatorial demand for resources, it's not clear these ideas could be used by practical resource-constrained computing systems.

### Clarity

This paper failed to communicate to its target audience. In the introduction, several key technical concepts are discussed without definition or sufficient context. One example is non-positively curved systems. Another is Gromov's Link Condition---an esoteric mathematical concept. If the authors want to see the adoption of these ideas in AI, then they need to recognize that average members of this audience will be unfamiliar with these concepts and require some motivation.

Instead of aiming at the entire AI community that uses grid worlds, it could be better to reorient this work toward a specific community, such as algorithmic game theory, reinforcement learning, or robotics. Ask how could these communities use your geometric characterization to enhance, simplify, or better understand their own algorithms. Right now it is not clear what anyone can do with this.

The paper claims it "relates to a growing body of research aimed towards understanding, from a geometric perspective, how deep learning methods transform input data into decisions, memories, or actions." However, this paper has little to do with deep learning. And furthermore, deep learning methods are usually applied in much more complex environments with many states.

Many concepts are precisely defined while others are vague. One example comes from the claim that determining collisions with this approach applies to multiple kinds of actions: "actions and inter-actions." It is unclear what an inter-action is in a grid world. Typically actions are represented as edges in the graph. Another example is "fog-of-war", which seems to relate to some notion of partial observability---a concept which seems unrelated to the paper's narrative around geometry and collision checking.

### Empirical Study

Simply put, there is no empirical study. Though I believe the paper could have designed small-scale experiments to demonstrate the utility of its geometric domain characterization. Consider experiments that support claims about collision checking properties in relation to existing collision checking algorithms.

### Potential Related Work

Consider how this work could be connected with prior work in robotics. For instance, Robert Ghrist had some work with Subrajit Bhattacharya on topological motion planning, which used knowledge of homotopy classes to perform efficient optimal planning. Work from this same group includes Jason Derenick, who uses simplicial approximation algorithms to efficiently solve common robot map building problems.

**Strengths And Weaknesses:**

### Strengths
* The presented mathematical ideas are potentially novel to AI.
* Collision checking is used in several areas of AI, and new insights could be impactful.
* Many potential connections to areas of AI research.


### Weaknesses
* Significance unclear and limited. Though the paper's goal was to introduce novel mathematical ideas to AI, it's not clear that this addresses a specific problem of significance.
* Empirical study is severely incomplete. The paper has an "Experiments and applications" section containing no experiments: no description of methodology, empirical research questions, or results.
* Poorly organized and unclear writing. The paper does not effectively communicate ideas to its intended target audience.

---

> ### Author Response · Authors · 2022-08-01
> **Response to Reviewer 2SSW**
>
> **Significance**
>
> The paragraph on conceptualising collisions is incorrect. Collisions are related to the *local geometry* of our modified state complexes (as detected by Gromov's Link Condition), not the topology. The topology captures the relative braiding of the agents, and is unrelated to collision-checking.
>
> Furthermore, the strange topology (arising even in empty space) is a limitation of the original Abrams-Ghrist-Peterson (AGP) setup. The purpose of our modifications is to remove this unnatural topological behaviour, thereby overcoming these limitations (L201-211). Consequently, our modified state complexes are better suited to serving as discrete analogues of configuration spaces compared to the original AGP version. Our modified state complexes offer insights into such domains comparable to those which can arise from using continuous configuration spaces.
>
> **Questions**
>
> State complexes are discrete objects which can capture all the states of a reconfigurable system (as introduced by AGP). In the context of agent-only gridworlds, state complexes can be viewed as discrete analogues of the classical configuration spaces (which are continuous objects); see L22-30.
>
> Collision-checking (a local problem) only requires the local geometry (via Gromov’s Link Condition). As such, no knowledge of the full state space topology is needed. However, for problems which are 'global' in nature (eg. route-planning), we must take the topology (eg. path homotopy classes) into account.
>
> **Audience**
>
> Regarding the suggestion to aim this work towards a specific sub-community, we believe this work has “many potential connections to areas of AI research”, as recognised as a strength by 2SSW. Opportunities for applications include, but are not limited to: analysing whether learning algorithms implement types of geometrically-inspired algorithms which perform so efficiently on cube complexes, or approximations thereof; whether learning algorithms construct aspects or features of cubical geometry, e.g., in hidden state representations, or whether they show improved performance if they are guided/constrained to using such geometry (in the spirit of [NZL20, ZZ22]); whether biological representations of allocentric and egocentric navigation (especially social place cells, i.e., dual encodings of place and conspecifics) reflect similar geometric properties; and to enhance data-efficient RL systems (as transition graphs improved performance [JZK+22]).
>
> **Writing**
>
> We structured our paper like so:
> 1. General introduction and summary of main contributions
> 2. Background on state complexes and our application to gridworlds
> 3. Computing/visualising state complexes of gridworlds, with simple examples to aid intuition
> 4. Topological limitations of AGP’s set-up and our modifications which address them
> 5. Main theoretical results relating collision-checking to local geometry
> 6. Outline of experiments, our open Python tool, and areas for application
> 7. Conclusion and discussion
>
> As some mathematical concepts may be unfamiliar to the AI community, we feel this is a sensible way to present the narrative. Nevertheless, it is quite jarring to receive two very different appraisals regarding the clarity and quality of presentation (see txoD’s comments).
>
> Moving the full technical definitions of Gromov's Link Condition and NPC cube complexes to the introduction would, we believe, significantly harm readability. We instead highlight some implications and uses of these tools/objects in the introduction and provide technical details in Appendix A1, including illustrated examples.
>
> **Clarity**
>
> By "actions and inter-actions", we mean things like discussed on L356-357. Our mentioning of "fog of war" was not in relation to partial observability (L287-292 and L303-305), but rather that a “4 step line of sight is sufficient” (as acknowledged by txoD).
>
> **Potential related work**
>
> There are certainly connections between our work and other applications of topology to fields such as robotics. When restricting to a specific path homotopy class, motion planning problems are simplified since we may instead work in the universal cover. (This follows from standard results in topology, rather than anything specific to robotics.) We expect existing greedy algorithms for NPC cube complexes to work reasonably well in the universal cover, with detours taken near potential collisions (L339-344). Interestingly, the universal cover is an infinite space, yet Bhattacharya and Ghrist were able to implement a practical algorithm.
>
> The topology of our state complexes captures dynamic features (relative motions of agents), whereas the topology of the simplicial approximations of Derenick et al. only captures static obstacles in the environment.
>
> **Additional references**
>
> [JZK+22] Zhengyao Jiang, Tianjun Zhang, Robert Kirk, Tim Rocktäschel, Edward Grefenstette. Graph Backup: Data Efficient Backup Exploiting Markovian Transition. arXiv:2205.15824 (2022).

---

### Author Response · Authors · 2022-08-01
**General comments to all reviewers**

We thank all reviewers for providing their assessments. Here we address general themes relevant to multiple reviews. We address further specific points in individual reviewer responses.


**Geometric perspective**
Our main goal is to establish a theoretical foundation for applications and further theoretical extensions whereby intrinsic geometric features in the task domain can be related to or directly detect important/relevant task information. Such geometric features needn't be learnt, they are simply directly available in the geometry, which we show an example of in collision detection and positive curvature (see Theorem 5.2). However, this framework is by no means limited to simple movements of agents or collision detection (L352-363).


**Computational considerations**
There is a misconception that when working with systems with a very large, perhaps infinite, number of possibilities that computational approaches to solving problems would be intractable. If this were the case, then doing arithmetic on a computer would be considered impractical as there are infinitely many numbers, hence infinitely many possible sums to consider. We don't expect any practical implementation of addition or multiplication to involve generating the entire times-tables. Nor, in our case, do we recommend constructing the entire state complex in practical applications (L316-318). Instead, and by our theorem, we only need to compute a small neighbourhood of states when needed in practice (L323-331).


Many examples where algorithms can be effectively implemented in situations involving infinite search spaces arise from decision problems in group theory. The Conjugacy Problem for a group $G$ asks: Given two elements $a,b \in G$, determine whether there exists an element $c \in G$ such that $ac = cb$. A famous result is that the Conjugacy Problem for braid groups can be solved in polynomial time [BKL98,FG03]. This does not involve naively searching the entire braid group for solutions (which is impossible since it is infinite). Instead, one exploits the underlying structure of the groups, using theoretical results to guarantee that if a solution exists, then it can be found using a bounded amount of work. In other words, you only need to generate enough data relevant to a given instance of the problem.


This approach of computing only what is necessary, only when necessary, is a fundamental principle behind lazy evaluation. A key feature of lazy evaluation is its ability to handle potentially infinite data structures. This is exemplified by the topological software package *Bigger* [Bel21], which is capable of doing computations on surfaces of infinite topological type (i.e. having infinitely many handles or holes). Thus, even for problems or systems that are intrinsically infinite, it is still possible to implement them so long as one can find suitable data structures (though this requires some careful thought and insight).


These examples demonstrate that an intractably large (or infinite) system or state space, in itself, poses no inherent obstacle to implementing practical algorithms. What is crucial, however, are insights into the system, geometric or otherwise.


**Empirical studies/comparisons to baselines**
2SSW and 5vs3 claim we did not do any or sufficient empirical studies. We never set out to do a large empirical study and our main contributions are theoretical. We did, however, do limited empirical studies for expository purposes and to showcase some of the potential benefits of this geometric way of thinking in future practical applications or theoretical extensions; see Appendix A4.


We are excited to see how state complexes will be incorporated into learning algorithms and ML pipelines for testing on relevant baselines. However, the current paper is the first to introduce state complexes to the AI community and its main contribution is in showing how purely geometric properties (which are not learned but are, rather, intrinsic to the task space) themselves detect important task features. We illustrated the example of collision detection and positive curvature, however we lay the foundation and offer many suggestions for extensions. With this in mind, we developed an open Python tool for researchers to construct state complexes of gridworlds for extensions and applications. Attempting further extensions and applications in this paper is beyond the scope and not particularly straightforward (see section "State complexes vs. transition graphs" in our response to 5vs3).


**Additional References**

[Bel21] Mark Bell. bigger (Computer Software). pypi.python.org/pypi/bigger, 2021. Version 0.3.1.

[BKL98] Joan Birman, Ki Hyoung Ko, Sang Jin Lee. A New Approach to the Word and Conjugacy Problems in the Braid Groups. Adv Math 139, 322-353 (1998).

[FG03] Nuno Franco, Juan González-Meneses. Conjugacy problem for braid groups and Garside groups. J Algebra 266, 112-132 (2003).

---

> ### Author Response · Authors · 2022-08-09
> **Further general comments to AC and reviewers**
>
> (As of current writing, we have yet to receive any further comments to our replies to the reviewers.)
>
> The reviews received thus far from 2SSW and 5vs3 contain severe factual errors and misunderstandings. They do not provide a meaningful assessment of our work.
>
> **2SSW**
>
> 2SSW has misjudged the significance of our work due to severe misunderstandings (see their ‘Significance’ section, and their confusion regarding the basic differences between geometry and topology). Crucially, they missed the central point of our paper.
>
> 2SSW has mistaken the ‘strange topology’ as a limitation of our modified state complexes, when in actual fact, it is a limitation of the original Abrams-Ghrist-Peterson setup. This is a critical misjudgement: we have explicitly stated in our paper (see the abstract, introduction, and Section 4) that the purpose of our modifications was to overcome these topological limitations. Indeed, txoD correctly points out that this is the main contribution of our paper, rather than the specific application to collision-checking. It is disappointing that 2SSW has completely missed this contribution beyond the state-of-the-art.
>
> 2SSW makes some references to applied topology work. However, as we explain in our reply to 2SSW (under ‘Potential related work’), these works are fundamentally different to our own, further suggesting conceptual misunderstandings.
>
> Ironically, 2SSW raises concerns about the computational feasibility of working with intractably large objects, yet suggests that we consider connections to using ‘knowledge of homotopy classes’ as in the work of Bhattacharya and Ghrist on robotics. The benefit of using homotopy classes is that one can instead work with the ‘universal cover’ - this simplifies the topology, at the cost of working with an infinite object. So the Bhattacharya-Ghrist motion-planning algorithm is a demonstration that one can implement practical algorithms on an *infinite* object! It is not clear if 2SSW even understands the underlying mathematics of Bhattacharya-Ghrist to realise that it actually counters their concerns regarding computational feasibility. (We are not sure if they are aware of lazy evaluation for that matter - see ‘Computational considerations’ under General Comments.)
>
> 2SSW again contradicts themselves in a less subtle way: by suggesting to aim this work towards a specific sub-community, while simultaneously claiming a strength of our paper is that it has ‘many potential connections to areas of AI research’.
>
> 2SSW claims we did not do empirical work. This is false – we did so, and this is acknowledged by both txoD and 5vs3. As stated in the paper, we conducted these experiments for purely expository purposes, not to provide any numerical comparisons with baselines. For a comparison to SOTA, we again point out that our modified state complexes do not suffer from the topological limitations that exist with AGP’s setup in the context of gridworlds; see:
>
> |      |  Agent braiding captured by topology  | Potential collisions detected by local geometry |
> |:----:|:-------------------------------------:|:-----------------------------------------------:|
> |  AGP state complex | Partially (topological defects arise) |                        No                       |
> | Our modified state complex |                **Yes**                |                     **Yes**                     |
>
> Given our efforts in gearing the exposition towards a non-specialist audience (as acknowledged by txoD), we find it particularly jarring to receive comments about the ‘poor organization and unclear writing’. It is also unclear what they mean by ‘unconvincing empirical remarks’ given that we supposedly did no empirical study. In our opinion, a score of 1 for Presentation should only be reserved for manuscripts that are clearly in an early draft stage and riddled with errors/typos. This is certainly not the case for our paper.
> All-in-all, the review of 2SSW is misguided and confused at best.
>
> **5vs3**
>
> 5vs3 demonstrates almost no basic understanding of the paper. They mistakenly equate state complexes with transition graphs (see our reply to 5vs3) and, based on this misjudgement, proceed to completely dismiss our work. They exhibit no understanding of the underlying geometry and topology, or our contribution beyond the work of AGP. On this unjustified basis they assert or imply false claims, e.g., ‘identification of unsafe states by observing *transitions* between states can be *exactly* captured by Q-learning, *which propagates values to implicitly identify the geometry of the state complexes*’ (emphasis added).
>
> We find it surprising that 5vs3 claims to have the highest confidence in their review, yet demonstrates the least understanding.

---

> > ### Comment · Reviewer_2SSW · 2022-08-09
> > **Response**
> >
> > I have read the authors' response and the other reviews, and found these unconvincing. Although the authors' response clarified a small technical point, their response seems focused on undercutting my review by mischaracterizing it. Furthermore, the other reviews did not convince me that my concerns were invalid. Therefore, I still believe this paper should be rejected.
> >
> > To reiterate, my main concerns are as follows.
> > 1. Significance unclear and limited: Failed to communicate how these ideas could be operationalized in AI. Computational tractability unclear.
> > 2. Empirical study is severely incomplete: Establishes no support for paper's claims that fall outside the scope of theory. As the authors put it "we conducted these experiments for purely expository purposes."
> > 3. Poorly organized and unclear writing: Unsupported claims. Sloppy with language, jargon, and delineation of technical concepts.
> >
> >
> > I have provided some detailed responses to points the authors made.
> >
> > > 2SSW has mistaken the ‘strange topology’ as a limitation of our modified state complexes, when in actual fact, it is a limitation of the original Abrams-Ghrist-Peterson setup.
> >
> > Thank you for clarifying this. My other concerns still remain.
> >
> > > Ironically, 2SSW raises concerns about the computational feasibility of working with intractably large objects, yet suggests that we consider connections to using ‘knowledge of homotopy classes’ as in the work of Bhattacharya and Ghrist on robotics.
> >
> > This is a mischaracterization. The potential related work I suggested was intended to provide successful examples of how related mathematical concepts could be brought into new application domains.
> >
> > > 2SSW again contradicts themselves in a less subtle way: by suggesting to aim this work towards a specific sub-community, while simultaneously claiming a strength of our paper is that it has ‘many potential connections to areas of AI research’.
> >
> > Not a contradiction---changing the target audience would not negate this strength.
> >
> > Indeed, I think one of this paper's strengths is the potential for _its ideas_ to connect with many areas of AI. Though to make a significant contribution, I believe these ideas need to be operationalized, or the paper needs to articulate how to operationalize them. Unfortunately, the current paper fails to do this to the general AI audience. One way to possibly remedy this issue is to aim the work at smaller target audience and operationalize it for those researchers.
> >
> > > 2SSW claims we did not do empirical work.
> >
> > Correct, this paper contains no experiments that provide evidence in support or against an argument the paper makes.

---

> > > ### Author Response · Authors · 2022-08-10
> > > **Further response to 2SSW**
> > >
> > > We are surprised 2SSW finds our detailed responses and other reviews unconvincing. We also find it notable that 2SSW has avoided in the above reply engaging with any of the mathematical content, arguments, results, or references.
> > >
> > > 2SSW claims that we
> > >
> > > > clarified a small technical point, [and our] response seems focused on undercutting [their] review by mischaracterizing it’.
> > >
> > > Firstly, we did not clarify a small technical point - we clarified one of the main contributions of our paper which 2SSW misunderstood due to fundamental confusions between geometry and topology. Secondly, we have provided many detailed responses in good faith and criticised (we believe fairly) the severe falsities and misunderstandings demonstrated in reviews from both 2SSW and 5vs3.
> > >
> > > 2SSW reiterated that the following were their main concerns:
> > >
> > > > Significance unclear and limited: Failed to communicate how these ideas could be operationalized in AI. Computational tractability unclear.
> > >
> > > We communicated how these ideas could be operationalised, both in the paper (sections 1, 4, 5, and 6) and in our responses (see our response to 2SSW under ‘Audience’). 2SSW has ignored and not engaged with any of these parts of the paper or responses. Nor has 2SSW recognised the key information provided by our Theorem and how this was discussed in the paper (L326-331) as having a strong operational implication (in this example) for safe, efficient multi-agent navigation: “In practical applications, such as calculating collision-avoiding navigation routes, it is – again, by Theorem 5.2 – only necessary to construct a small local subcomplex.”
> > >
> > > Additionally, we have described in detail why the computational tractability concern is wrong (see ‘Computational considerations’ above), which again 2SSW has not engaged with. We even explained how the Bhattacharya-Ghrist work (which 2SSW referenced) undermines 2SSW’s point. 2SSW claims they referenced Bhattacharya-Ghrist because they intended to ‘provide successful examples of how related mathematical concepts could be brought into new application domains’. Indeed, Bhattacharya-Ghrist have successfully used ideas from topology to implement an algorithm on an object that is computationally intractable to fully construct. We would appreciate it if 2SSW could pinpoint exactly why similar approaches would fail in our case. Otherwise, their claim only serves to validate our previous remark: “It is not clear if 2SSW even understands the underlying mathematics of Bhattacharya-Ghrist to realise that it actually counters their concerns regarding computational feasibility.”
> > >
> > > > Empirical study is severely incomplete: Establishes no support for paper's claims that fall outside the scope of theory.
> > >
> > > The paper makes no claims that fall outside the scope of theory, so this point is moot. We even acknowledge the limitations of our theoretical study (L386-392): “we have so far only explored very simple AI environments. Further work is needed to expand the framework and results to more general, sophisticated, and real-world environments … we caution against hasty real-world implementation of the main results … it would still be important to perform rigorous checks and tests in application domains, since our results do not directly extend to situations beyond which the stated assumptions hold.”
> > >
> > > > Poorly organized and unclear writing: Unsupported claims. Sloppy with language, jargon, and delineation of technical concepts.
> > >
> > > Other than seeming to suggest we move Appendix A1 into the introduction (which we believe will harm readability, see ‘Writing’ in our response to 2SSW), 2SSW provided no examples to support their concerns regarding sloppy writing. They also have not identified any specific claim which they believe is unsupported. If 2SSW could point to specific instances, we would be happy to address these as we have done so for other questions/concerns. As it stands, these criticisms are just vague blanket statements which add nothing constructive to the discussion.
> > >
> > > 2SSW makes further comments:
> > >
> > > > Indeed, I think one of this paper's strengths is the potential for its ideas to connect with many areas of AI. Though to make a significant contribution, I believe these ideas need to be operationalized, or the paper needs to articulate how to operationalize them.
> > >
> > > Again, we have identified these both in the paper and in our responses. 2SSW, however, has ignored and not engaged in any of this content or the relevant responses.
> > >
> > > > this paper contains no experiments that provide evidence in support or against an argument the paper makes.
> > >
> > > On L340-341, we state that “heuristically, we expect most states to satisfy NPC, and so existing greedy algorithms for calculating geodesics will work well in most situations.” This is supported by the empirical evidence given in Appendix 4.

---

### Meta-Review · Area_Chair_c5GD · 2022-08-26

**Recommendation:** Reject
**Confidence:** Certain

**Metareview:**

This paper analyses grid worlds, more precisely multiple objects moving in grid worlds, using the mathematical idea of state complexes.  The state complex represents all possible configurations as a single space, from which domain properties can be ascertained by group-theoretic, combinatorial, or geometric analysis.  In particular, the paper develops a theory around "Gromov's Link Condition" to analyze conditions under which collisions can be prevented in such domains.

The reviewers had a mixed initial response to this paper.  On the positive side, the reviewers appreciated the theoretical development (txoD) and novelty (2SSW).  On the negative side, the reviewers struggled to see the significance or relevance of the work to learning or AI (2SSW, 5vs3).  The reviewers understood the work as a mechanism for collision checking (2SSW), a means to support learning (5vs3), and a computational mechanism for analyzing gridworld dynamics (txoD).  The author response clarified several aspects of the reviews that were misunderstood.  The author response did not sway the reviewers.  Primarily, the concern is that the paper failed to communicate the relevance of the mathematical analysis of gridworlds to an AI audience.  The sole positive reviewer ultimately concurred with the arguments made by the negative reviewers.

Two reviewers indicate to reject, and one indicates a weak accept.  Based on the failure of the paper to clearly communicate the relevance of its ideas to any reviewer, the paper is rejected.  One suggestion for a future revision would be to present these ideas in the general setting of an MDP (instead of a specific domain of a gridworld).  The local combinatorial analysis on a generic MDP could potentially be more useful to the MDP community when considering planning for AI safety or problems of mechanism design.  The evidence needed to validate the ideas for those communities might again be different from the evidence provided in this paper.  As a separate comment, the analysis of the transition dynamics of actions may have related work stemming from predictive state representations.  In the paper's current form, the reviewers were unable to see a clear contribution.

**Award:**

No

---

### Decision · Program_Chairs · 2022-09-14

Reject